# Degradation of Rhodamine B in Wastewater by Iron-Loaded Attapulgite Particle Heterogeneous Fenton Catalyst

**Peiguo Zhou *, Zongbiao Dai, Tianyu Lu, Xin Ru, Meshack Appiah Ofori**  **, Wenjing Yang, Jiaxin Hou and Hui Jin**

College of Biology and the Environment, Nanjing Forestry University, Nanjing 210037, China; dzongbiao@163.com (Z.D.); luty.snei@sinopec.com (T.L.); r1227877604@163.com (X.R.); appiahmeshack8@gmail.com (M.A.O.); yyywj7089@163.com (W.Y.); houjiaxin@163.com (J.H.); nmlx1993@163.com (H.J.)
* Correspondence: zhoupeiguo@njfu.edu.cn; Tel.: +86-13913390757

**Abstract:** The water pollution caused by industry emissions makes effluent treatment a serious matter that needs to be settled. Heterogeneous Fenton oxidation has been recognized as an effective means to degrade pollutants in water. Attapulgite can be used as a catalyst carrier because of its distinctive spatial crystal structure and surface ion exchange. In this study, iron ions were transported on attapulgite particles to generate an iron-supporting attapulgite particles catalyst. BET, EDS, SEM and XRD characterized the catalysts. The particle was used as a heterogeneous catalyst to degrade rhodamine B (RhB) dye in wastewater. The effects of $H_2O_2$ concentration, initial pH value, catalyst dosage and temperature on the degradation of dyes were studied. The results showed that the decolorization efficiency was consistently maintained after consecutive use of a granular catalyst five times, and the removal rate was more than 98%. The degradation and mineralization effect of cationic dyes by granular catalyst was better than that of anionic dyes. Hydroxyl radicals play a dominant role in RhB catalytic degradation. The dynamic change and mechanism of granular catalysts in catalytic degradation of RhB were analyzed. In this study, the application range of attapulgite was widened. The prepared granular catalyst was cheap, stable and efficient, and could be used to treat refractory organic wastewater.

**Keywords:** iron-bearing attapulgite; heterogeneous Fenton; granular catalyst; rhodamine B

## 1. Introduction

With the acceleration of the processes of urbanization and modernization, the economy continues to grow to meet the needs of humanity. Oil extraction, fine chemical industry, production of non-ferrous metals, pesticides, the pharmaceutical industry and cooking, printing and dyeing light industries, paper production, and other areas with high energy consumption and high pollution bring many conveniences to people's lives while causing significant damage to the environment and public health. For example, industrial wastewater often contains a lot of organic pollutants that are difficult to degrade, which not only causes serious environmental pollution but also threatens human health and safety. Therefore, governments worldwide attach great importance to it [1–3].

Among the many industrial water pollution environments, the pollution of printing and dyeing and textile wastewater to the environment is more serious. Wastewater contains many toxic and harmful pollutants that are difficult to be biodegraded [4–6]. Especially the dye wastewater has a great impact on human health and the natural environment. Rhodamine B is a highly water-soluble red dye with high toxicity. Rhodamine B is widely used in textiles, printing and dyeing, food, and other fields because of its low price. This dye is difficult to degrade in water and also causes various diseases, such as cancer [7]. At present, the treatment methods of dye wastewater include adsorption [8,9], biological treatment [10], membrane separation [11], solvent extraction [12], etc. Some treatment methods have drawbacks such as high cost, substandard water quality, and secondary

pollution, etc. Considering the advantages and disadvantages of the existing treatment methods of dye wastewater, developing a new and cost-effective approach is a matter of great concern.

Advanced oxidation technology is an innovative treatment method, which contains highly active oxidation free radicals and can effectively degrade different organic pollutants in wastewater [13–15]. To better degrade pollutants in water, advanced oxidation technology has implemented a variety of methods, such as electrochemical oxidation [16–18], ozone oxidation [19], photocatalytic oxidation [20], ultrasonic oxidation [21], catalytic wet oxidation [22], and Fenton oxidation [23]. Fenton oxidation is superior to other advanced oxidation technologies in the simplicity of reactions and operation, rapid and efficient degradation of target pollutants, low cost, and safety for the environment. Hydroxyl radicals produced by Fenton reaction destroy the structure of organic matter and mineralize it into carbon dioxide and water. However, the traditional homogeneous Fenton system has some problems, such as narrow reaction pH range, iron sludge generation, difficult recycling and catalyst reuse, which minimize its application. Researchers fixed $Fe^{2+}$ ($Fe^{3+}$) or iron oxides ($Fe^{II}$ and $Fe^{III}$) in the Fenton reagent on the support to form a heterogeneous Fenton system to overcome these drawbacks. Various materials have been studied and developed as heterogeneous Fenton catalyst supports, such as molecular sieve, enriched alumina, silica, mineral clay, transition metal, and iron oxide composites, etc. [24]. Researchers fabricated a heterogeneous catalyst system supported by iron on a modified molecular sieve to treat non-degradable materials in molasses distillation wastewater. Studies showed that the decolorization rate was 90% and the TOC removal rate was 60% after using the sulfuric acid alternative catalyst [25]. Hernandez-Olono et al. studied the degradation of rhodamine B, methyl orange and methylene blue by precipitation on alumina under UV irradiation [26]. It was found that the degradation rates of Rhodamine B, methyl orange and methylene blue on $Fe/Al_2O_3$ dried under $H_2O_2$ during UV irradiation were 99.6%, 100%, and 99.1%, respectively. Using roasted red clay as the iron source, a heterogeneous light-Fenton method was constructed to remove C.I. Acid Red 17 (AR17), and the study showed that decolorization efficiency was as high as 94.71% [27]. Hu et al. used four kinds of containing mesoporous silica materials for catalytic treatment of chlorophenol wastewater. They found that Cu-doped SBA-15 with Cu/Si mole ratio of 0.133 had the best catalytic activity for oxidative degradation of 4-chlorophenol. The apparent reaction rate constant was 0.170 $min^{-1}$, which was 1.3~3.6 times that of other Cu-containing catalysts [28]. Chen et al. prepared $Mn-Fe_3O_4/RGO$ composite as a multiphase photo-Fenton catalyst for degradation of Rhodamine B (RhB) [29]. Under natural pH conditions, the low dosage of catalyst was 0.2 g/L within 80 min, and the degradation rate reached 96.4%. When pH was 2 and 11, the removal rates were 91% and 85%, respectively. After 10 cycles, the catalyst still maintained a high degradation efficiency of about 90%. In addition to the above materials, many scholars have also developed bimetal composite materials as heterogeneous Fenton catalysts to degrade organic pollutants [30–33]. However, these catalysts are complex, expensive, rare and require high cost, which may lead to secondary pollution. Thus, their application in the catalytic advanced oxidation process is limited. Therefore, developing low-cost, environmentally friendly and naturally available heterogeneous Fenton catalysts for water treatment is desirable.

Attapulgite, as a common clay mineral, is abundant in yield and low in price. It also has a large specific surface area, unique spatial crystal structure and surface ion-exchange property. In addition, attapulgite has a good adsorption capacity for organic pollutants in water, which can be used as a cheap natural adsorbent for research in the field of wastewater treatment [34,35]. The rupture of the Si-O bond forms the electronegativity of Si-OH and the loss of coordination water in attapulgite. Therefore, positively coordinated metal ions in promulgating are usually used as catalysts. Cao et al. supported CuO particles on attapulgite and studied the catalytic oxidation of CO by the CuO/ATP catalyst at a low temperature, obtaining a catalytic effect similar to that of metal oxide-supported CuO [36]. Wang et al. used precipitation, impregnation and mechanical blending methods to load

Ni on attapulgite to generate a Ni/ATP catalyst for bio-oil hydrogen production [37]. The results showed that the catalytic hydrogen production rate of Ni/ATP prepared by the precipitation method reached 82%, and the conversion rate of acetic acid reached 85%, which was considerably lower than the Ni/ATP catalyst prepared by the impregnation method. Ma et al. supported platinum nanocatalysts with HCl acidified attapulgite and the results showed remarkable catalytic activity and selectivity for p-chloronitrobenzene, m-chloronitrobenzene, and o-chloronitrobenzene [38].

In this work, iron ions were supported on attapulgite particles and used as heterogeneous Fenton catalysts to degrade RhB dye in wastewater. The basic properties of the granular catalyst were studied. At the same time, the effects of initial concentration, dosage, initial pH value, temperature, and $H_2O_2$ concentration on the catalytic degradation effect were investigated to find the best catalytic reaction conditions. Under the optimum catalytic reaction conditions, the reusability, stability and degradation effect of different dyes of granular catalysts was studied. Finally, the dynamic change of catalytic degradation of dye RhB by a granular catalyst was studied, the formation and reaction position of a reactive free radical were determined, and the catalytic degradation mechanism of a granular catalyst was analyzed and speculated upon.

## 2. Results and Discussion

### 2.1. Catalyst Characterization

2.1.1. BET and EDS Analysis of Catalyst

The chemical elements of the granular catalyst were analyzed. As shown in Table 1, the main elements of the catalyst were Si and O. This was consistent with the XRD pattern. The mass percentage of iron in the catalyst was 12.09%. EDS analysis showed that there was iron in the granular catalyst, which led to the Fenton reaction.

**Table 1.** EDS spectrograms of the Fe-loaded attapulgite particles and distribution of elements.

| Element | O | Mg | Al | Si | K | Ti | Fe | the Total |
|---|---|---|---|---|---|---|---|---|
| Weight percentage (wt.%) | 54.44 | 2.3 | 4.9 | 24.42 | 1.21 | 0.63 | 12.09 | 100 |

Figure 1a shows the catalyst's nitrogen adsorption and desorption isotherms, which were type IV isotherms [39], indicating that the internal pores of the catalyst were mainly mesoporous, with a BET-specific surface area of 119.5409 $m^2$/g and an average pore size of 11.1603 nm. The catalyst had good adsorption performance.

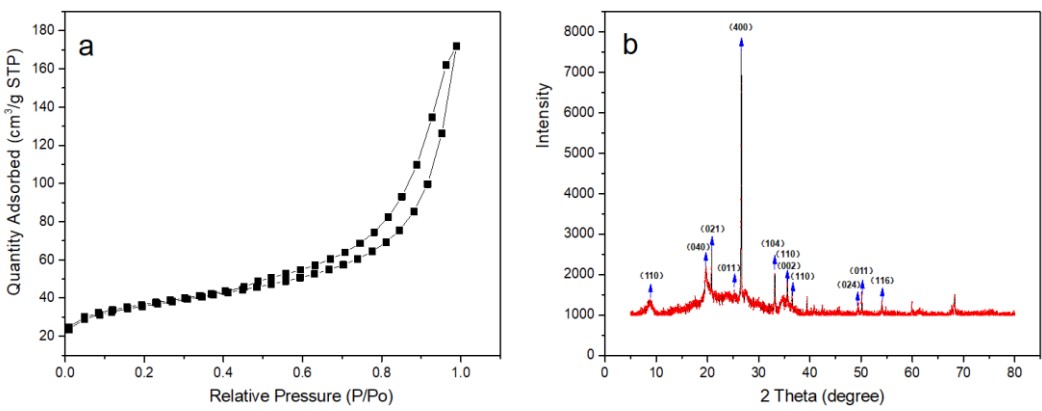

**Figure 1.** $N_2$ adsorption/desorption isotherm of granular catalyst at 77 K (**a**) and XRD (**b**).

Figure 1b XRD spectra of iron-loaded catalysts shows that attapulgite characteristic diffraction peaks appeared at $2\theta = 8.5°, 19.8°, 20.3°, 27.9°, 35.7°$, etc. The corresponding crystal plane spacing was D (110) = 10.3974 nm, D (040) = 4.4725 nm, D (021) = 4.3829 nm,

D (400) = 3.1942 nm, D (002) = 2.5139 nm, respectively. The $SiO_2$ characteristic diffraction peaks appeared at 2θ = 26.7°, 36.6° and 50.2°, and the corresponding crystal plane distances were D (011) = 3.3421 nm, D (110) = 2.4560 nm, D (011) = 1.8171 nm, respectively. In addition, α-$Fe_2O_3$ cubic spinel characteristic diffraction peaks were found at 2θ = 33.2°, 35.6°, 49.5°, 54.1°, etc. The corresponding crystal plane spacing was D (104) = 2.6989 nm, D (110) = 2.5171 nm, D (024) = 1.8408 nm, D (116) = 1.6943 nm, but the overall peak strength was much weaker than $Fe_2O_3$ crystal. This could be because only a small part of $Fe(NO_3)_3$ was converted into the α-$Fe_2O_3$ crystal in the calcination process of the iron-loaded catalyst, while most of the remaining $Fe(NO_3)_3$ could be converted into other crystal iron oxides or only attached to the catalyst surface in the form of some chemical bonds without showing the corresponding crystal structure.

### 2.1.2. SEM Analysis of Catalyst

Analysis of the particle catalyst before and after the reaction with scanning electron microscopy (SEM) can be seen in Figure 2; Figure 2b shows the particle catalyst after reaction, and Figure 2a shows the reaction front of the catalyst particles. It can be seen that the internal structure has been greatly damaged, the rod between the crystal is an organic whole repeatedly, and the initial needle bar gradually formed a similar layered structure, but most of the channels remain and the catalyst can be recycled many times.

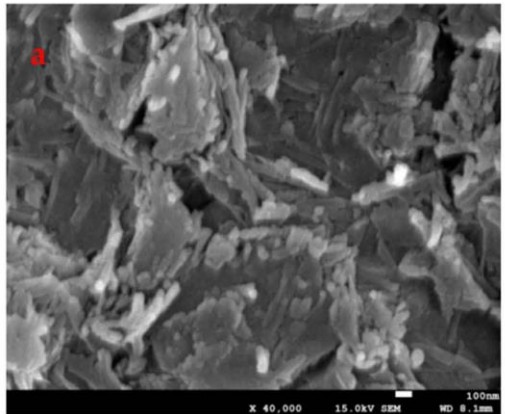 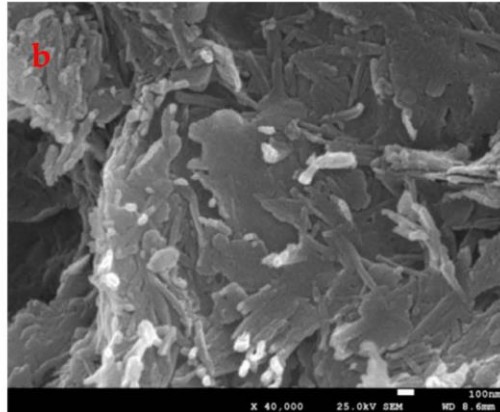

**Figure 2.** SEM image of catalyst before (**a**) and after (**b**) reaction (×40,000).

### 2.2. Influence of Reaction Conditions on Degradation Performance

### 2.2.1. Influence of Different Reaction Systems on RhB Degradation

It can be seen from Figure 3a,b that in different reaction systems, the degradation and mineralization effects of RhB were not ideal when a hydrogen peroxide solution was added only at 35 °C and pH 3. After 240 min, the removal rate of RhB and the TOC mineralization rate were only 5.98% and 0.92%, respectively. When only a granular catalyst was added to the reaction solution without $H_2O_2$, the concentration of RhB and TOC decreased in the same trend, indicating that the reduced RhB concentration was only adsorbed by the granular catalyst, no molecular ring-opening or degradation mineralization occurred, and the adsorption amount reached 39.61% at 240 min. When 0.5 mg/L $Fe^{3+}$ and 196 mmol/L $H_2O_2$ solutions were added to the reaction solution to form a homogeneous Fenton system, the homogeneous Fenton reaction caused by a small amount of dissolved $Fe^{3+}$ insignificantly contributed to the catalytic degradation of RhB in the system regarding the RhB removal rate and mineralization effect. Therefore, it can be seen that the degradation of dye-RhB by an iron-loaded attapulgite granular catalyst was mainly contributed to by a heterogeneous Fenton reaction, which conformed to the heterogeneous surface catalytic mechanism. When the particle catalyst was added to join the $H_2O_2$ in the reaction solution, which constituted the similar Fenton system, the reaction temperature and solution pH value also affected their degradation performance. With the increase of temperature, the

molecular energy increases and the movement is violent, the amount of active free radical increases, and the oxidative degradation rate is accelerated. At the same time, the influence of pH on the degradation performance was different: in terms of decolorization effect, it was alkaline > acidic > neutral, and in terms of mineralization effect, it was acidic > neutral > alkaline.

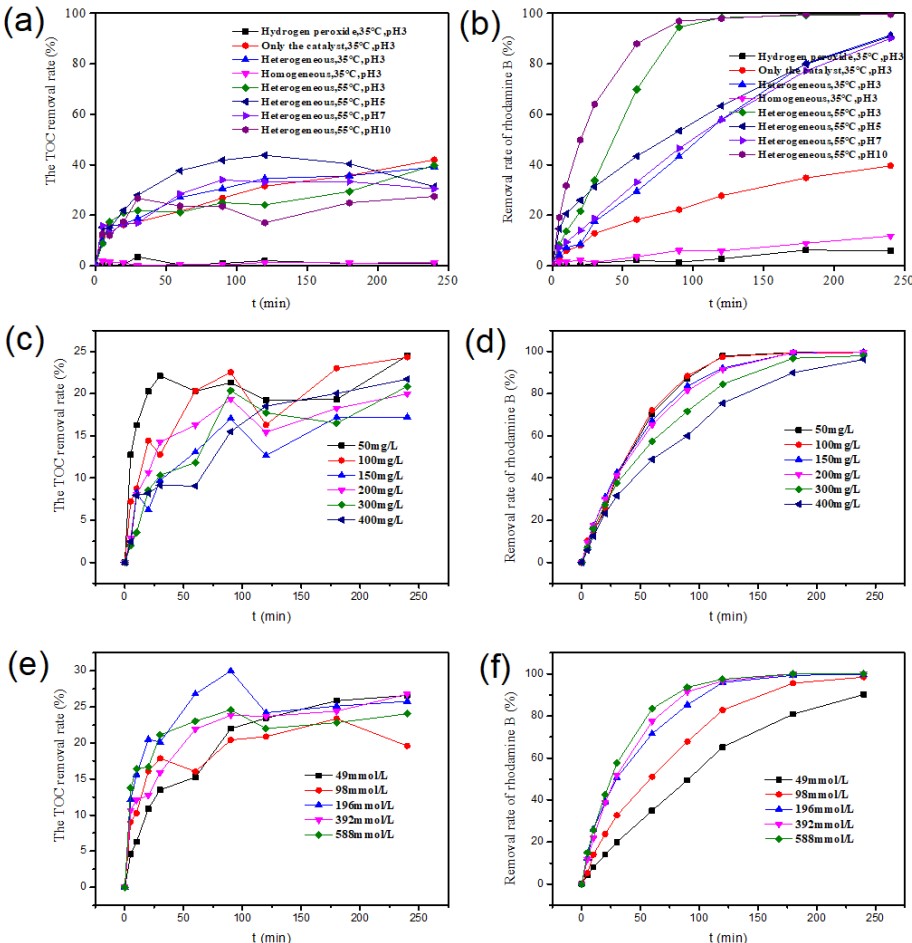

**Figure 3.** Effects of different reaction systems (**a,b**) ($c_0$ = 100 mg/L, $c_{H_2O_2}$ = 196 mmol/L, catalyst = 10 g/L); (**c,d**) RhB initial concentration ($c_{H_2O_2}$ = 196 mmol/L, catalyst = 10 g/L, pH = 10, T = 55 °C); (**e,f**) $H_2O_2$ concentration ($c_0$ = 200 mg/L, catalyst = 10 g/L, pH = 10, T = 55 °C); on degradation and mineralization rate of Rhodamine B.

### 2.2.2. Influence of RhB Initial Concentration

Figure 3c,d shows the effect of initial dye concentration on RhB dye degradation. It can be seen that the degradation and decolorization reaction of RhB took place before the mineralization reaction, indicating that the decolorization of the dye was easier than the mineralization, and the lower the initial concentration, the faster the degradation and mineralization rate. When the initial concentration was 50 mg/L, the removal rate of RhB was close to 90% after 90 min, the decolorization was almost complete after 120 min, and the removal rate of TOC reached 24.52% after 240 min. With the increase of RhB's initial concentration, the decolorization and mineralization rates gradually decreased. The removal rate of 400 mg/L RhB solution reached 90% after 180 min, and the TOC removal rate was 21.73% after 240 min. This could be because the higher the concentration of RhB, the more molecules per unit volume. In addition, the generation of reactive free radicals was constant, so the degradation and mineralization rate of RhB decreased with the increase in concentration [40]. Considering the obvious degradation effect, the RhB solution with an initial concentration of 200 mg/L was selected for subsequent experiments.

2.2.3. Influence of Different $H_2O_2$ Concentrations

In the heterogeneous Fenton system, the concentration of hydrogen peroxide affected the amount of active free radicals [41], and the decolorization and mineralization efficiency of RhB dye in the solution. As can be seen from Figure 3e,f, when the concentration of $H_2O_2$ in the reaction system increased from 49 mmol/L to 588 mmol/L, the oxidation decolorization and mineralization rate of the RhB solution increased gradually. After 240 min of reaction, the degradation rate of RhB increased from 90.17% to 99.99%. It is verified that the RhB solution can be decolorized to a certain extent by adding a low concentration of $H_2O_2$ in the heterogeneous Fenton system, while the removal rate of TOC does not increase significantly and is stable at about 25%. Meanwhile, it also indicates that the free radicals generated in the alkaline reaction system only destroy the chromophore group of the RhB molecule. Therefore, dye decolorization took precedence over mineralization because only a small part of it can eventually be mineralized into $CO_2$ and $H_2O$. When the concentration of $H_2O_2$ in the reaction system increased to 196 mmol/L, the decolorization rate reached 95.92% after 120 min, which greatly shortened the reaction time and accelerated the decolorization rate compared with a lower $H_2O_2$ concentration. The RhB removal rate reached 99.77% after 240 min. This could be because the reaction's amount of reactive free radicals increased with an increase in $H_2O_2$ concentration, more free radicals attacked the RhB molecule, and the removal rate of RhB and TOC further improved [42]. After that, increasing the concentration of $H_2O_2$ in the reaction system had little contribution to the oxidative degradation and mineralization effect of RhB, and excessive $H_2O_2$ in the solution could lead to a self-annihilation reaction or a series of ineffective reactions with the active free radicals generated in the catalytic reaction process [43,44]. As a free radical scavenger, active oxidation substances originally used to attack RhB molecules in the reaction system were consumed, affecting oxidative degradation [45,46]. On the other hand, when the amount of catalyst in the reaction system was fixed and the concentration of $H_2O_2$ was properly increased, the number of collisions between the $H_2O_2$ molecule and catalyst increased per unit time and the reaction rate accelerated. Due to the limited amount of catalyst, the collision number of $H_2O_2$ concentration did not increase any further, which was macroscopically expressed as the RhB removal rate slowing down. Therefore, considering the oxidation decolorization and degradation of RhB, an appropriate $H_2O_2$ concentration of 196 mmol/L was selected under alkaline conditions.

2.2.4. Influence of Different Catalyst Dosages

As can be seen from Figure 4a,b, under the conditions of the above appropriate initial concentration, pH value and hydrogen peroxide concentration, the degradation and mineralization rate of RhB was significantly accelerated with the increase in catalyst dosage. The removal rate of RhB was only 78.81% after 120 min in the system with a catalyst dosage of 2 g/L. At the same time, the removal rate of RhB in the system with the dosage of 20 g/L reached 98.73%, which was nearly 20% higher. The dosage of the catalyst was positively correlated with the catalytic efficiency of the reaction system to a large extent, indicating that the addition of catalysts provided more adsorption sites on the one hand, which could absorb more RhB molecules. The removal rate of RhB and TOC was improved. On the other hand, the increase in catalyst dosage could provide more reactive sites to a greater extent, and the rate and number of oxidative active free radicals generated by the catalyst and $H_2O_2$ reaction are greatly improved [47,48], which increased the degradation efficiency of the reaction system. When the amount of catalyst was low, the reaction rate of the heterogeneous Fenton system was slow. With greater dosage, the catalytic reaction is inhibited to a certain extent, and the occurrence of side reactions is promoted, affecting the rate of oxidative degradation. As can be seen from the TOC removal rate, when the catalyst dosage increased, the TOC removal rate in the reaction system increased first and then decreased, because the greater the dosage of the catalysts, the greater the adsorption of RhB molecule in the system at the beginning, and the higher the TOC removal rate. As the reaction progressed, the adsorbed RhB molecule was gradually oxidized and degraded

into small molecules by active free radicals. However, the particle catalyst prepared in this experiment had a weak adsorption capacity for intermediate products of these small molecules, so it was desorbed from the pore channel of the particle catalyst to the liquid solution, which showed a temporary decrease in TOC removal rate. Subsequently, these small molecule intermediates were further mineralized and degraded into $CO_2$ and $H_2O$, so the TOC removal rate increased again. From an economic perspective, the appropriate catalyst dosage for subsequent experiments was 10 g/L.

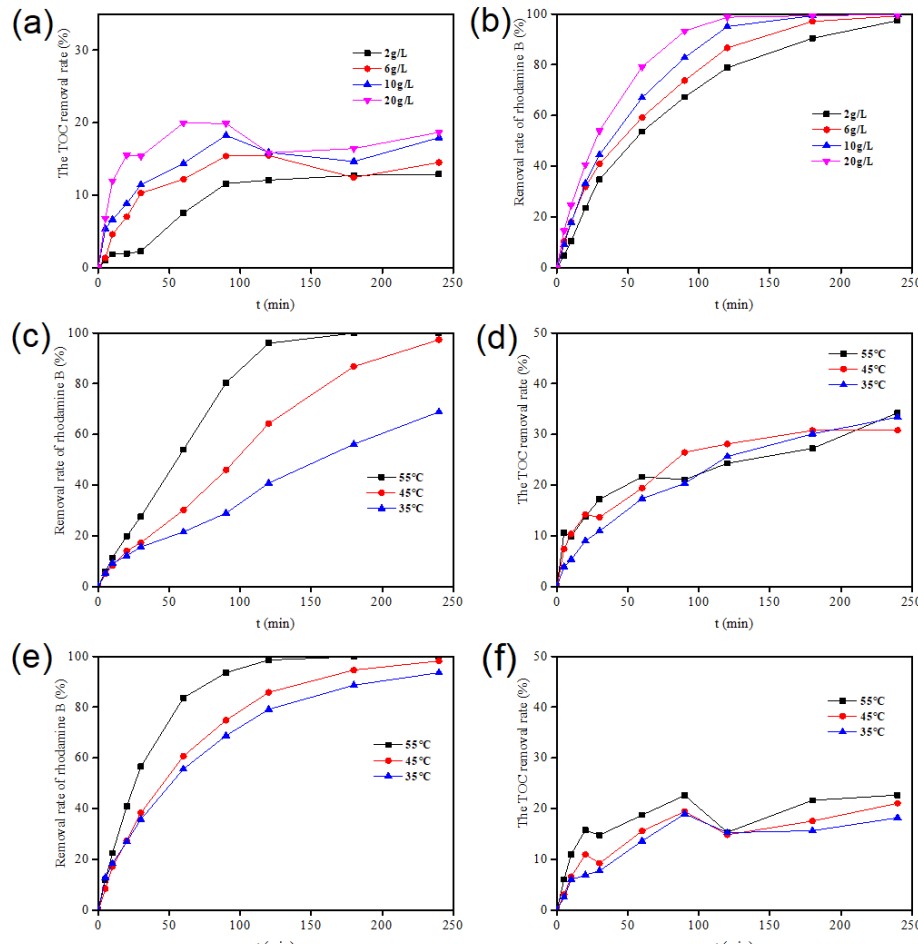

**Figure 4.** Effects of different reaction temperature (**c,d**) ($c_0$ = 200 mg/L, $c_{H_2O_2}$ = 196 mmol/L, pH = 3, catalyst = 10 g/L), (**e,f**) ($c_0$ = 200 mg/L, $c_{H_2O_2}$ = 196 mmol/L, pH = 10, catalyst = 10 g/L); catalyst dosage (**a,b**) ($c_0$ = 200 mg/L, $c_{H_2O_2}$ = 196 mmol/L, pH = 10, T = 55 °C) on the removal rate of rhodamine B.

#### 2.2.5. Influence of Initial pH Value

In a heterogeneous Fenton reaction system, the change of pH greatly influences the oxidation decolorization and mineralized degradation of dyes [41,49]. As can be seen from Figure 5, in terms of the RhB decoloring effect, it was alkaline condition > acid > neutral conditions, and in terms of the RhB mineralization effect, it was the acid condition > neutral condition > alkaline conditions. According to the literature, the application of the pH range in the mentioned conventional Fenton system has a very big distinction [50].

Under neutral conditions, it can be seen that pH had a negligible effect on the oxidation, decolorization, and degradation of RhB, and the decolorization rate was very slow. The decolorization rate only reached approximately 50% after 120 min of reaction, and the oxidation degradation effect was relatively weak due to the adsorption of the RhB molecule in the solution by a particle catalyst. Under alkaline conditions, with the increase in

pH, the decolorization rate of RhB by the granular catalyst increased gradually, and the decolorization rate reached more than 99% in the same reaction time. However, the TOC removal rate was lower than that under neutral and acidic conditions, and the TOC removal rate was about 25%. The results showed that only a small part of RhB molecules adsorbed by the granular catalyst could be eventually mineralized into $CO_2$ and $H_2O$. Under the acid condition, the oxidation of RhB was also very quick, and the decolorization rate was improved with the reduction of pH. Finally, with the initial pH value of 3 and a reaction time of 120 min, the decolorization rate reached 96%, which was far higher than that of the neutral condition. Compared to neutral conditions and acidic conditions, the removal rate of TOC experienced a period of growth, and the process of further decrease and increase was consistent with the rule mentioned in the previous sections, which also indicated that the granular catalyst first adsorbed the RhB molecule, then degraded into small molecule intermediates, was desorbed, and finally was mineralized into $CO_2$ and $H_2O$, which conformed to the heterogeneous surface catalysis mechanism.

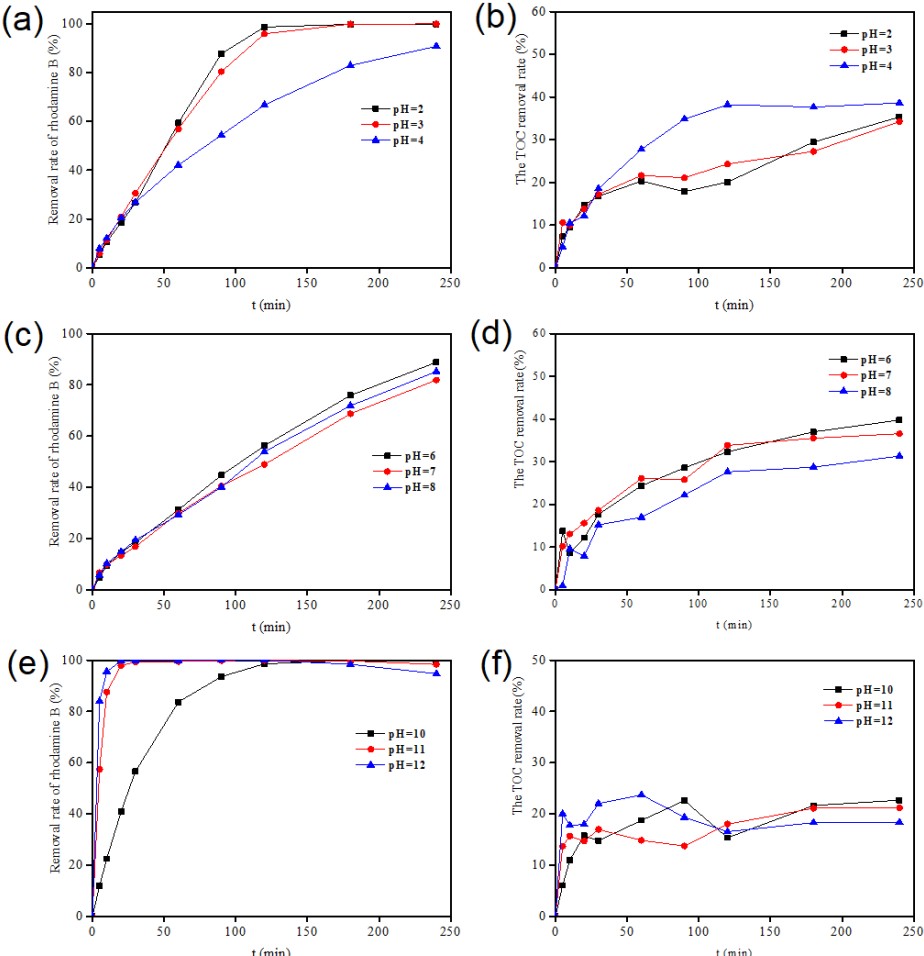

**Figure 5.** Effects of different pH values on the removal rate and mineralization rate of rhodamine B ($c_0$ = 200 mg/L, catalyst = 10 g/L, T = 55 °C, $c_{H_2O_2}$ = 196 mmol/L).

On the whole, it can be seen that the decolorization effect of RhB was better in alkaline conditions than in acidic conditions. The mineralization effect of dye was better in acidic conditions than in alkaline conditions because $H_2O_2$ was relatively stable in acidic conditions. At the same time, $H_2O_2$ was easy to decompose into peroxide radical (•OOH) and $H^+$ in alkaline conditions. Under the condition of the constant addition of the granular catalyst and hydrogen peroxide in the system, the rate and number of free radicals generated under acidic conditions were low, while the rate and number of

free radicals generated mainly by ●OH were high under alkaline conditions, mainly by ●OOH. The strong oxidation capacity of ●OH can further mineralize the small molecule intermediates into $CO_2$ and $H_2O$, while ●OOH is difficult to further mineralize these intermediates. Therefore, the mineralization effect under alkaline conditions is inferior to that under acidic conditions. However, the pH of the reaction solution should not be too high or too low. Moreover, excessive acid and alkali environments destroyed the structure of the granular catalyst, resulting in particle dissolution and iron ion loss. On the other hand, the concentration of ●OH was inversely proportional to the concentration of $[OH^-]$, and the formation of ●OH was inhibited when the pH of the reaction solution was too low [51]. The conversion balance between $Fe^{III}$ and $Fe^{II}$ on the catalyst surface was destroyed, affecting the catalytic reaction [52], and the specific reaction mechanism remains to be explored. Nevertheless, the particle catalyst prepared in this study solved the problem of pH limitation of the traditional homogeneous or heterogeneous system very well [53] and still had a good decolorization effect under alkaline conditions, which was conducive to the extensive application of the heterogeneous Fenton system.

### 2.2.6. Influence of Temperature on Degradation of Rhodamine B

The effect of Fenton-like oxidation reaction on decolorization efficiency and mineralization of 200 mg/L RhB dye solution at different temperatures was studied. From the acidic (c) and (d) groups with pH 3 and alkaline (e) and (f) groups with pH 10 in Figure 4, the catalyst dosage was 10 g/L and $H_2O_2$ concentration remained unchanged at 196 mmol/L. As can be seen from Figure 5, regardless of the system, with the increase in temperature, the oxidation degradation rate was accelerated, RhB removal rate and the removal rate of TOC increased. When pH was 3 and reaction temperature was 35 °C, the reaction rate of RhB had an obvious rise after fall, and this trend remained constant. This was because the particle catalyst only had an adsorption effect on RhB at the beginning, or the adsorption rate was much higher than the oxidative degradation rate. With the progress of the reaction, a large amount of $H_2O_2$ entered into the iron oxide inside the particle catalyst and on the surface of the particle catalyst to form a heterogeneous Fenton system and generate active free radicals. The RhB molecules adsorbed on the pore surface were oxidized and degraded into small molecules that fall off from the surface, and then continued to adsorb RhB molecules in the solution, forming a cycle of adsorption–degradation–desorption–re-adsorption–re-degradation–re-desorption. At that time, the oxidation degradation process was dominant, so the reaction rate in the system increased twice. The energy was required for the formation of active free radicals, so the higher the temperature in the system, the higher the energy [54], the greater the probability of breaking the O-O $H_2O_2$ bond and the higher the concentration of active free radicals, which significantly accelerated the process of oxidative degradation [55]. Therefore, there was no secondary increase in the reaction rate when the reaction temperature was 55 °C. In actual industrial applications such as printing and dyeing, the wastewater discharge temperature is higher, so the reaction temperature of 55 °C was chosen for the follow-up experiment.

### 2.3. *Study on Sustainability and Stability of the Granular Catalyst*

It is an ideal research target for heterogeneous Fenton catalysts with certain regeneration, reuse and good stability. The particle catalyst reached a saturation state after 360 min. Therefore, $H_2O_2$ was added after the particle catalyst was adsorbed for 360 min to investigate the change in the concentration of the RhB dye under the optimal catalytic reaction conditions. As can be seen from Figure 6, only when a granular catalyst was presented in the solution, the RhB residual rate was 50.26% after 360 min. After that, with the increase in adsorption time, the concentration of the RhB dye did not decrease much. At 480 min, the residual rate of RhB was still as high as 47.07%, which indicated that the adsorption of RhB on the granular catalyst reached saturation. Therefore, a certain amount of $H_2O_2$ was added immediately after the particle catalyst was adsorbed and saturated. It can be seen that under both acidic and alkaline conditions, the concentration of RhB in the solution



decreased rapidly, and the residual rate of RhB was less than 1% after 120 min, indicating that after adding $H_2O_2$, the particle catalyst could effectively degrade and completely decolorize the RhB molecules adsorbed on the surface of particle channels, so the prepared particle catalyst had a certain in situ regeneration performance.

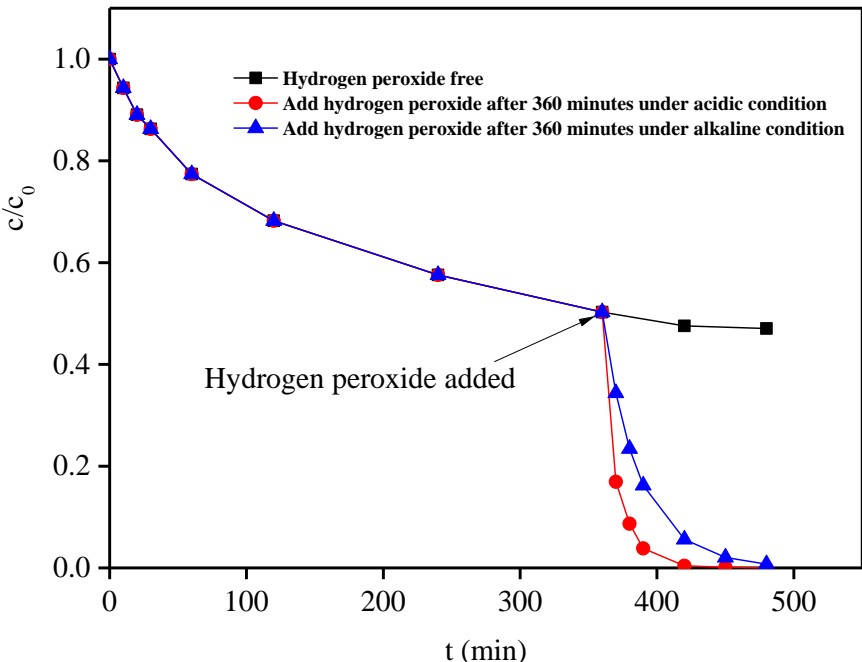

**Figure 6.** Concentration change of $H_2O_2$ on Rhodamine B after adsorption equilibrium of granular catalyst.

In practical engineering applications, the catalyst itself must be reusable and have an excellent stable structure [56,57], so the reusability of the annular catalyst is also an important evaluation index. The granular catalyst was put into a 100 mL RhB solution with an initial concentration of 200 mg/L; the catalyst dosage was kept at 10 g/L, the $H_2O_2$ concentration was 196 mmol/L, the pH value was 3 and 10, and the solution temperature was 55 °C. The reaction was repeated five times, the granular catalyst was removed and washed then dried with deionized water. As can be seen from Figure 7a,b, RhB can be effectively decolorized and degraded by a granular catalyst 180 min after five repeated reactions under acidic and alkaline conditions. Decoloring performance remained the same, and the basic removal rate was stable and constituted over 98%, which showed that the preparation of catalysts had good reusability. However, the mineralization effect of RhB decreased. After the first use of granular catalyst, the mineralization rate of RhB reached 28.11% (pH = 3) and 25.75% (pH = 10). After repeated reuse, the mineralization rate decreased. The mineralization rate of the fifth use was only 22.16% (pH = 3) and 16.28% (pH = 10), as shown in Figure 7c. This was due to the dissolution of a small amount of infirm bound iron ions on the catalyst's surface after each repeated reuse, as shown in Figure 7d. Another reason was that a small amount of organic matter in the previous reaction was adsorbed and remained on the surface of the catalyst channel, resulting in the reduction of active sites and the TOC removal rate.

### 2.4. Study on the Degradation Effect of Different Dyes by the Granular Catalyst

The above results showed that the granular catalyst had a significant degradation effect on RhB. The feasibility of its application in dye wastewater can be better clarified through the degradation experiment of different dyes. Methylene blue and Congo red were used as the target pollutants to degrade as RhB in the same reaction system. Figure 8 showed that under alkaline conditions, the reaction rate of cationic RhB was faster than that of the acidic conditions, and the TOC removal rate in the acidic conditions was higher than

that of alkaline conditions. The other cationic dye (methylene blue) also showed a similar trend; the difference was a good effect on the mineralization of the methylene blue on rhodamine B. For cationic dyes, the alkaline condition of the heterogeneous Fenton system was conducive to the oxidation and decolorization of dyes, while the acidic condition was conducive to the degradation and mineralization of dyes. For anionic dyes like Congo red, the decolorization and mineralization effects were worse than those of cationic dyes, which was because the surface of ferric attapulgite granular catalyst prepared in this experiment was partially replaced by $Al^{3+}$ with $Si^{4+}$ in the 4-valence coordinates, and by $Mg^{2+}$ with $Al^{6+}$ in the 6-times coordinates, resulting in electronegativity on the surface of the granular catalyst. Therefore, it has a certain repulsive effect on anionic dyes.

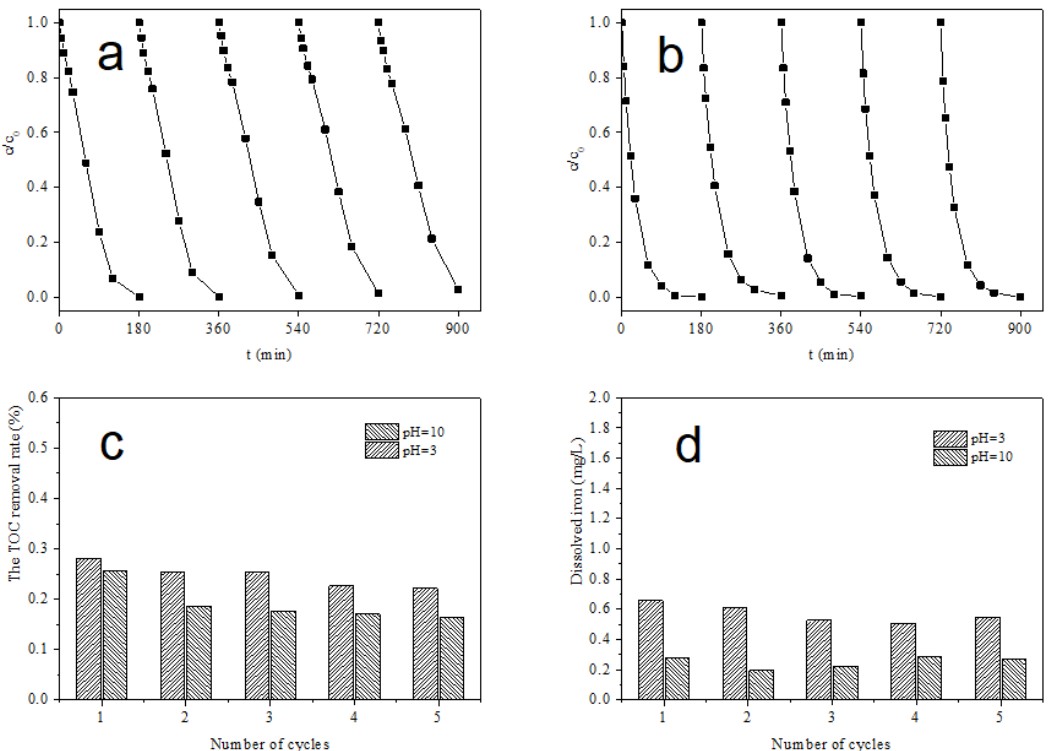

**Figure 7.** Reusability of the granular catalyst under acidic (**a**) and alkaline (**b**) conditions and mineralization rate of Rhodamine B (**c**) and iron dissolution amount (**d**).

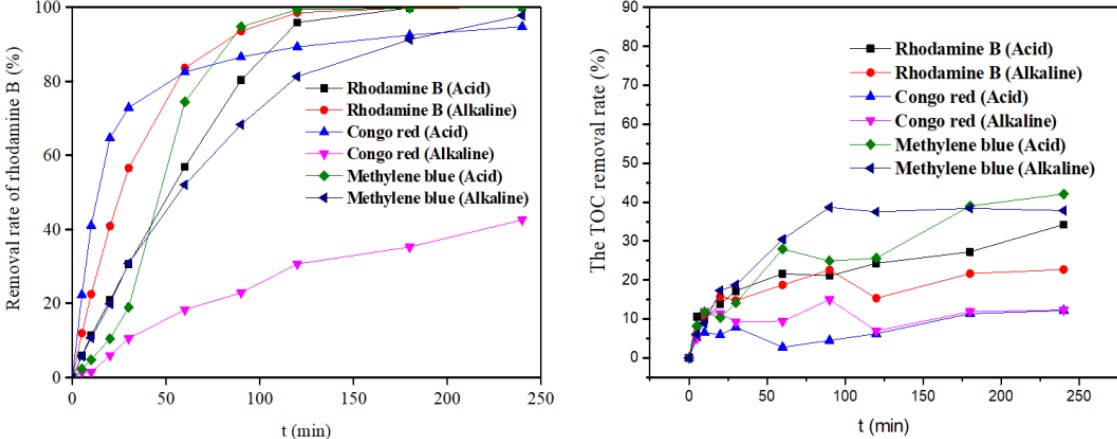

**Figure 8.** Degradation and mineralization of different dyes ($c_0$ = 200 mg/L, $c_{H_2O_2}$ = 196 mmol/L, T = 55 °C, catalyst = 10 g/L).

### 2.5. Dynamic Study on the Catalytic Degradation Process of the Granular Catalyst

2.5.1. Dynamic UV-VIS Spectra of RhB during the Reaction Process

The degradation process of RhB was described qualitatively by UV-VIS spectroscopy. Under the optimal reaction conditions, samples were taken at different times for dilution and placed in a quartz colorimetric dish for full-wavelength scanning within the wavelength range of 250–800 nm, as shown in Figure 9. It can be seen from the figure that the RhB solution had the maximum characteristic absorption peak at 554 nm in the visible region, which was mainly caused by the benzene amino group, carbonyl group, and four ethyl groups in RhB molecular structure [58]. According to relevant literature, with the progress of the reaction process, the degradation of RhB molecule mainly went through n-site diethyl, destruction of the large conjugated structure, ring-opening and mineralization, etc. The macroscopic manifestation was the gradual fading of dye color, and the absorption peak at 554 nm of the UV-VIS spectrum in the figure also gradually weakened. At 180 min, it can be seen that the absorption peak almost disappeared, indicating that the oxidation decolorization of RhB was basically completed and decomposed into small molecule intermediates or completely mineralized into $CO_2$ and $H_2O$. This was due to the similar activity of the Fenton process for the production of oxygen free radicals, which directly attacked the molecular structure of the RhB conjugate with a large variety of anthracene (i.e., benzene amine and carbonyl), which led to its cracking and loss of characteristic absorption peak. At the same time, the hydroxylation effect led to the redshift of the absorption peak, and the N-ethyl process caused a blue shift in its absorption peak [59]. Therefore, it can be concluded that the degradation process of the RhB molecule in this experiment is the simultaneous destruction of n-site diethyl and large conjugated oxanthracene structures, and the red-shift and blue-shift effects cancel each other out, thus only showing a decrease in absorbance.

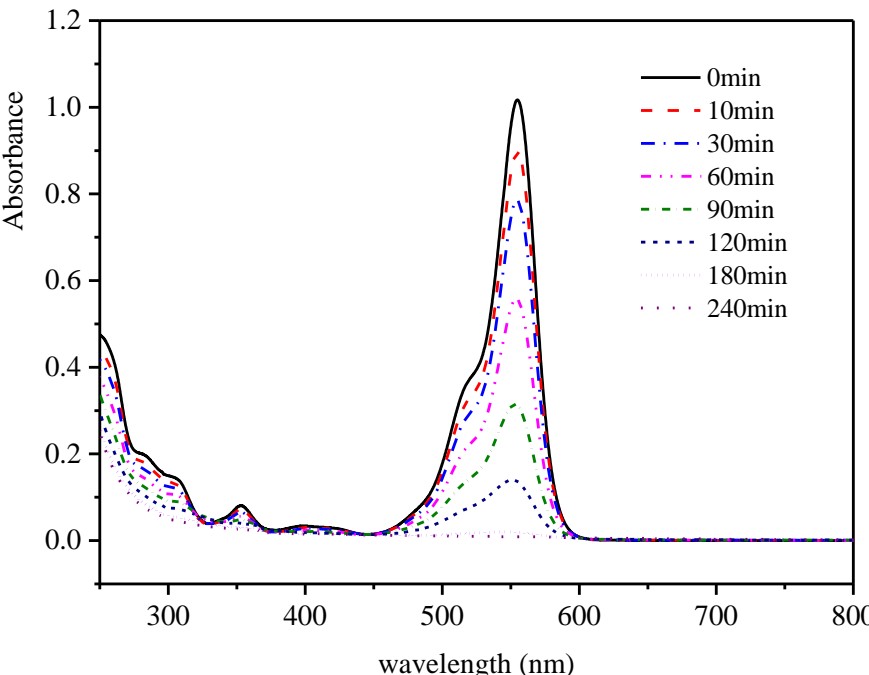

**Figure 9.** UV-vis spectra of RhB degradation ($c_0$ = 200 mg/L, $c_{H_2O_2}$ = 196 mmol/L, T = 55 °C, catalyst = 10 g/L, pH = 3).

2.5.2. Determination of Active Substances in the Reaction Process

In order to investigate whether the prepared granular catalyst was a free radical reaction in catalytic oxidation of RhB, the free radical trapping agent ascorbic acid was added in the reaction process to determine the presence of free radicals in the catalytic system according to the degradation of RhB [60]. Active substances that can catalyze

reactions in heterogeneous Fenton systems mainly include hydroxyl radical (●OH) [61], peroxide radical (●OOH) [62], and high-valent iron (Fe$^{IV}$) [63]. To confirm the types of free radicals playing a leading role in the heterogeneous Fenton system, different concentrations of tert-butanol (TBA) and potassium iodide were added as hydroxyl radical trapping agents under the optimal reaction conditions [64–66] to determine the existence of hydroxyl radicals in the reaction process. At the same time, a certain concentration of p-benzoquinone was added to the reaction solution to determine whether there were peroxy radicals in the reaction process.

As can be seen from Figure 10a, the addition of ascorbic acid led to a gradual decline in the degradation rate of RhB. When the amount of ascorbic acid was 2.5 mmol/L, the residual rate of RhB was almost equal to the adsorption rate of the catalyst, which preliminarily proved that the heterogeneous Fenton reaction system was a free radical reaction. As can be seen from Figure 10b,c, the catalytic degradation rate of the reaction solution with tert-butanol was reduced, indicating that a hydroxyl radical existed in the reaction solution and participated in the catalytic degradation process. However, as the concentration of tert-butanol increased, the degradation of RhB was not completely inhibited. The degradation rate was still 74.97% after 240 min, indicating that there were other active substances in the reaction system. After adding 0.2 mmol/L potassium iodide, the degradation rate of RhB decreased, and the degradation rate of RhB was 79.68% after 180 min. Potassium iodide captured hydroxyl radicals on the surface of the granular catalyst, but its inhibition was relatively small compared with tert-butanol, indicating that the granular catalyst catalyzed the degradation of RhB. The RhB molecules were first adsorbed on the surface of the granular catalyst and then degraded, while hydroxyl radicals catalyzed a small part of RhB molecules in the liquid phase. Figure 10d showed that under the acid condition, the degradation of RhB inhibition was limited, indicating that the heterogeneous Fenton system produced a very little amount of oxygen free radicals, from which it can be concluded that the hydroxyl free radical reaction in the process of catalytic degradation of RhB was dominant.

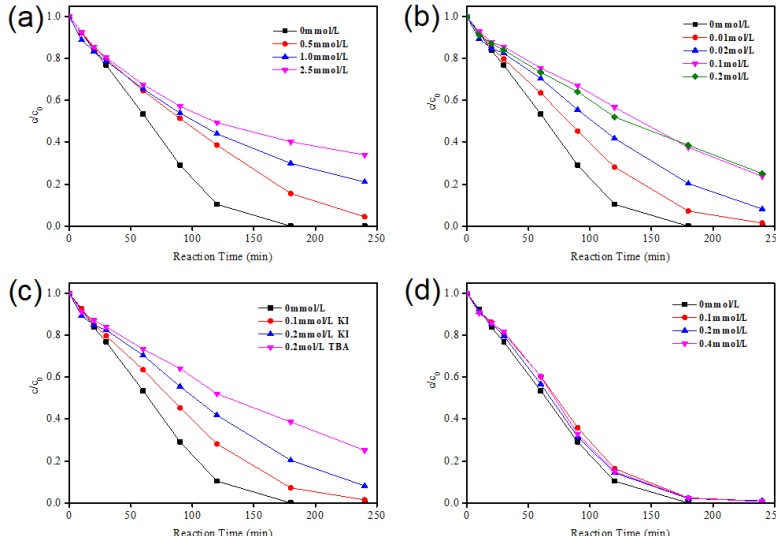

**Figure 10.** Effects of different types of free radical trapping agents on degradation of rhodamine B: (**a**) Ascorbic acid; (**b**) TBA; (**c**) KI and TBA; (**d**) P-benzoquinone($c_0$ = 200 mg/L, $c_{H_2O_2}$ = 196 mmol/L, T = 55 °C, catalyst = 10 g/L, pH = 3).

### 2.5.3. Formation Rule of Hydroxyl Radical (●OH)

According to the above analysis, hydroxyl radical reaction plays a dominant role in catalytic degradation of RhB. To investigate the rule of hydroxyl free radical, a certain concentration of coumarin solution was used as the trapping agent of the hydroxyl free radical, and the pH of the solution was 3 under the optimal reaction conditions. The

influence on the rule of hydroxyl free radical generation was investigated by changing the dosage of the catalyst and the hydrogen peroxide concentration.

Figure 11a,b shows that with an increase in the particle catalyst dosage, the generation of free radicals with similar heterogeneity in the Fenton system gradually increased, and $H_2O_2$ consumption also increased. This could be caused by the increase in the dosage of the particle catalyst which brought more adsorption sites and its reactivity [47] and increased the adsorption quantity of the catalyst. In addition, it increased the contact area of the catalytic reaction. $H_2O_2$ was in contact with more iron oxides, and its decomposition rate was accelerated, thus increasing the production of hydroxyl radical.

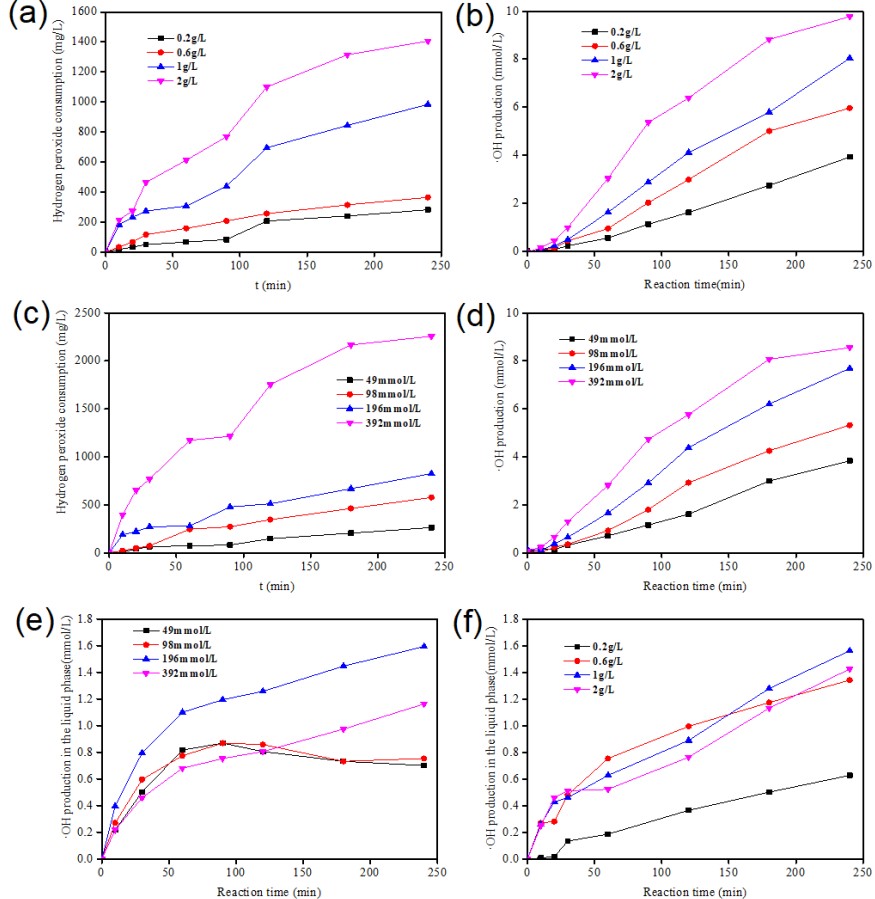

**Figure 11.** Formation rule of hydroxyl radical under different conditions: (**a**,**c**) The consumption of $H_2O_2$; (**b**,**d**) ●OH production; (**e**) $H_2O_2$ concentration (T = 55 °C, catalyst = 10 g/L, pH = 3); (**f**) catalyst dosage ($c_{H_2O_2}$ = 196 mmol/L, T = 55 °C, pH = 3).

Figure 11c,d showed that with the increase in $H_2O_2$ concentration in the solution, the production of hydroxyl radical in the heterogeneous Fenton system also gradually increased, which could be due to the increase in $H_2O_2$ concentration under the condition of maintaining the same number of adsorption sites and reactive sites. As a result, more $H_2O_2$ could enter the particle catalyst and contact with the iron oxide on the pore surface, increasing the contact frequency (the number of collisions per unit time) and resulting in a faster decomposition rate of $H_2O_2$ and production of hydroxyl radicals [67]. However, the generation rule of liquid hydroxyl radicals in the system was slightly different from that of total hydroxyl radicals, as shown in Figure 11e,f. It can be seen that the production amount of liquid hydroxyl radicals was much lower than that of total hydroxyl radicals, indicating that the reaction system was dominated by surface catalytic degradation reaction, supplemented by the liquid radical reaction. With the addition of the catalyst, the amount and rate of formation of liquid hydroxyl radical increased obviously. When the addition of

granular catalyst was 1 g/L, the amount of formation of liquid hydroxyl radical reached the maximum. After the addition of the catalyst, the formation rate of liquid hydroxyl radical did not increase. When $H_2O_2$ concentration was low, the generation rate of hydroxyl radical increased first and then decreased. With the increase in $H_2O_2$ concentration, hydroxyl radical generation in the liquid phase increased first and then decreased, indicating that the higher the $H_2O_2$ concentration, the better. A high concentration of $H_2O_2$ will lead to the greatest quenching of the hydroxyl radical generated in the liquid phase, reducing the reaction efficiency.

### 2.6. Speculation on the Mechanism of Catalytic Oxidation of RhB Dye by the Granular Catalyst

The degradation mechanism of the heterogeneous Fenton system mainly includes the following three kinds: heterogeneous surface catalysis mechanism, homogeneous catalysis mechanism of iron ion dissolution, and catalytic oxidation mechanism of high iron [68]. The heterogeneous surface catalysis mechanism is as follows: when irradiated with visible light, the dye is excited to reduce part of iron oxide $Fe^{III}$ on the catalyst's surface to $Fe^{II}$, and the catalyst adsorbs the RhB dye and $H_2O_2$ on the surface of the granular catalyst. The $H_2O_2$ adsorbed on the surface of the granular catalyst reacts with $Fe^{II}$ and $Fe^{III}$ on the surface to generate active free radicals. The RhB molecules adsorbed on the catalyst's surface are oxidized and degraded into small intermediate products or partially mineralized into $CO_2$ and $H_2O$. The products are desorbed from the particle surface, and then continue to adsorb RhB molecules in the solution, forming a cycle of adsorption–degradation–desorption–re-adsorption–re-degradation–re-desorption, until RhB is completely degraded. The oxidative degradation process is dominant. The homogeneous catalytic mechanism of iron ion dissolution is as follows: the heterogeneous Fenton particle catalyst of ferric attapulgite dissolves a small number of iron ions during the reaction process, which forms a homogeneous Fenton system with $H_2O_2$ in the solution, catalyzes the decomposition of $H_2O_2$ to generate active free radicals, and degrades RhB. According to the previous experimental results, the Fenton reaction in this part of dissolved iron ions insignificantly contributes to the degradation of RhB in the system, which is a secondary path of degradation reaction. The mechanism of catalytic oxidation with high iron is as follows: during the conversion of $Fe^{II}$ and $Fe^{III}$ on the catalyst surface, some iron oxides are converted into higher $Fe^{IV}$, and high iron directly oxidizes RhB to small molecular intermediates or partially mineralizes into $CO_2$ and $H_2O$ [69]. This study took attapulgite particles as the supported and loaded iron ions on the surface of attapulgite to make a heterogeneous granular catalyst, which was used for the degradation of the RhB dye. The mechanism of the degradation process was shown in Figure 12.

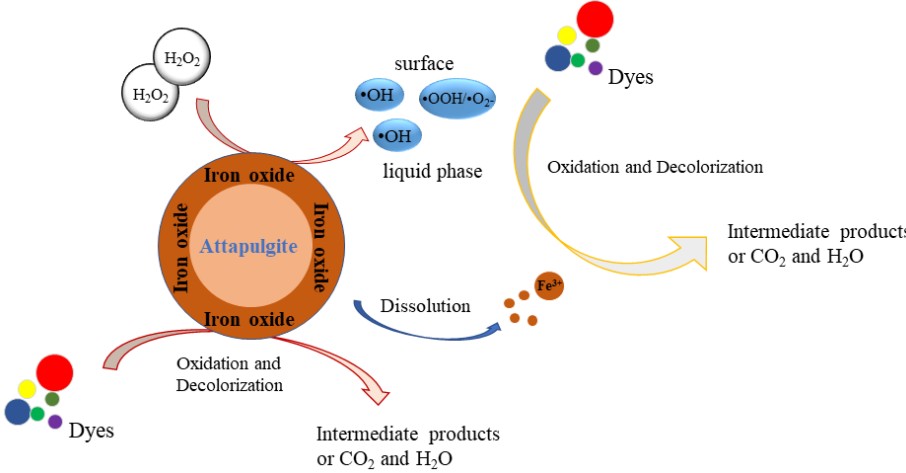

**Figure 12.** Proposed catalytic mechanism of dye decolorization and oxidation in the system.

### 3. Experimental Materials and Methods

*3.1. Reagents and Materials*

Rhodamine B (RhB; $C_{28}H_{31}ClN_2O_3$) was provided by the Tianjin Institute of Chemical Reagents (Tianjin, China). Congo red (CR; $C_{32}H_{22}N_6Na_2O_6S_2$) was purchased from Shanghai Maclin Biochemical Technology Co., Ltd. (Shanghai, China). Methylene blue (MB; $C_{16}H_{20}ClN_3S$) was provided by Sinopharm Chemical Reagents (Shanghai, China). Sodium hydroxide (NaOH), 30% hydrogen peroxide ($H_2O_2$), nitric acid ($HNO_3$), hydrochloric acid (HCI), potassium dichromate ($K_2Cr_2O_7$), concentrated sulfuric acid ($H_2SO_4$), and mercury sulfate ($Ag_2SO_4$) were purchased from Nanjing chemical reagents (Nanjing, China). Iron trioxide ($Fe_2O_3$), iron nitrate ($Fe(NO_3)_3 \cdot 9H_2O$), p-benzoquinone ($C_6H_4O_2$), potassium iodide (KI), ascorbic acid ($C_6H_8O_6$), and silver sulfate were provided by Sinopagol Chemical reagents (Nanjing, China). Tert-butyl alcohol ($C_4H_{10}O$) was purchased from Shanghai Lingfeng Chemical reagent (Shanghai, China). Coumarin ($C_9H_6O_2$) was purchased from Aladdin (Shanghai, China). 7-hydroxycoumarin ($C_9H_6O_3$) and potassium titanium oxalate ($K_2TiOC_4O_8 \cdot 2H_2O$) were purchased from McLean (Shanghai, China). All reagents used were analytically pure. Attapulgite (ATP) was found in Xuyi, Jiangsu Province. Attapulgite was prepared into granular supported iron ions as the catalyst.

*3.2. Preparation and Characterization of Catalyst*

Different additives and attapulgite were mixed evenly in a certain proportion, and a certain quality of deionized water was added into a molded shape for mixing and rubbed into 1–3 mm particles. The mixture was placed in the oven at 110 °C to dry, and then into the tube furnace to calcinate for a certain time. The prepared particles were immersed in a solution of iron ion concentration of 1.6 mol/L, the ratio of solid to liquid was 20:1, the immersion temperature was 80 °C, and the immersion time was 4 h. After filtration and repeated washing several times, the obtained products were dried in a 110 °C oven and then calcined in a tubular furnace at a certain temperature to prepare an iron-bearing attapulgite granular catalyst.

The ASP-2020 specific surface area and pore size analyzer of Micromeritics were used to measure the specific surface area at the degassing station of the analyzer at a rate of 10 °C/min to 300 °C for 10 h at the temperature of liquid nitrogen (77 K). EDS analysis was carried out under the conditions of 40–50,000 times magnification and gold spraying on the surface of dry samples. Ultima IV multifunctional composite X-ray diffractometer θ = 5~80° (Cu target Kα, λ = 1.5406 A) was used to test the samples; the JCPDS file number was 87–2096. Jsm-7600f Thermal field emission scanning electron microscope (JEOL) was used to spray gold (Au) on the surface of the absolute dry samples with a magnification of 40~50,000 times.

*3.3. Instruments and Analytical Methods*

The absorbance of rhodamine B in the solution was determined by a TU-1810 UV-vis spectrophotometer at 554 nm and its concentration was analyzed. Meanwhile, the solution's Congo red and methylene blue can be determined by ultraviolet spectrophotometer and visible spectrophotometer at 497 nm and 664 nm. The pH of the solution was measured using an EOTECH pH 700 pH meter. A TOC-vcpn (Shimadzu, Japan) total organic carbon analyzer was used to determine the TOC value of the solution. The concentration of $Fe^{3+}$ in the reaction solution was determined by a TAS-990 Super atomic absorption spectrophotometer. The reaction of hydrogen peroxide with potassium titanium oxalate was carried out in an acetic acid-sodium acetate buffer solution to form a stable orange complex. The absorbance of this orange complex was measured at 375 nm and the concentration of hydrogen peroxide ($H_2O_2$) was analyzed.

### 4. Conclusions

This study of the use of natural clay minerals content load iron load attapulgite particles of iron catalyst was prepared by using XRD, EDS and BET methods to study the

basic properties of the catalyst particles itself, and found that particle catalysts for RhB dye not only have a catalytic degradation effect, but also has a certain adsorption performance and adhere to the surface of the catalyst particles in an iron oxide crystal shape. In the process of reaction, the amount of iron dissolved in the granular catalyst is very low. In the catalytic reaction process, conditions are changed to investigate their effects on the dye RhB catalytic degradation performance and find out the best reaction conditions. The study found that under the condition of alkaline dye, the decolorization rate is greater than the acid condition; the lower the initial concentration and reaction, the greater the concentration of $H_2O_2$ and catalyst dosing quantity, the higher the reaction temperature, and the greater and faster the dye RhB degradation and mineralization. However, pH value has a great influence on the degradation and mineralization of RhB. In terms of the decolorization effect, alkaline condition > acidic condition > neutral condition, and in terms of the mineralization effect, acidic condition > neutral condition > alkaline condition. Under the optimal catalytic reaction conditions, granular catalyst's reusability, stability and degradation effects on different types of dyes were studied. It was found that the prepared granular catalyst has a charge-negative surface, and the degradation and mineralization effects of cationic dyes are better than that of anionic dyes. It was found that hydroxyl radical reaction dominated the catalytic degradation of RhB dye in a heterogeneous Fenton system, in which surface radical reaction was the main catalyst and liquid radical reaction was the auxiliary one. The study shows a simple and environmentally friendly route that can be scaled up and seems to be a promising approach for industrial wastewater treatment in the future.

**Author Contributions:** P.Z.; methodology and review, Z.D.; disgining-analysis and writing, T.L. and X.R.; data curation, M.A.O.; writing—original draft preparation, W.Y., J.H. and H.J.; visualization and collection. All authors have read and agreed to the published version of the manuscript.

**Funding:** This research was funded by the National Key Research and Development Program "Key Technology of Safety Production and Pollution Monitoring of Wood-Based Panel" (2016YFD0600703).

**Institutional Review Board Statement:** Not applicable, this study did not involve human or animal studies.

**Informed Consent Statement:** Not applicable, this study did not involve human or animal studies.

**Data Availability Statement:** This study did not report any data.

**Conflicts of Interest:** The authors declare that they have no conflict of interest.

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
