# Peer review of "Degradation of Rhodamine B in Wastewater by Iron-Loaded Attapulgite Particle Heterogeneous Fenton Catalyst"

_catalysts, doi:10.3390/catal12060669_

Round 1

Reviewer 1 Report

Comments to the authors

The manuscript reports results concerning the preparation of iron-bearing attapulgite granular catalyst for the photodegradation of Rhodamine B dye. Designing an efficient degradation system for dye degradation have received increased attention in the last few decades. The present work affords a promising strategy for the designing effective and economical catalyst for the efficient degradation of Rhodamine B dye. I suggest major revision for quality enhancement, which are given bellow.

11.      The manuscript contains grammatical and typing mistakes which should be carefully read and remove and the language need improvement.

22.      Discuss some toxic effect of Rhodamine B dye in the introduction section of the manuscript.

33.      Provide the maximum absorbance of methylene blue and congo red dyes.

44.      Why Figure 1b is discussed before than 1a. Arrange in series.

55.      Provide JCDPF file no. to all the XRD of all the samples.

66.      Perform SEM and TEM characterization for surface analysis.

77.      Write catalysts sustainability instead of catalysts use.

88.      Label the corresponding peaks with crystal planes in the XRD figure.

99.      The authors should properly correlate the XRD data with the results presented in the manuscript.

110.  Explain the mechanism and affecting parameters with the help from these reviews and cite in the manuscript “Water 2022, 14, 242. https://doi.org/10.3390/w14020242 and Journal of Environmental Chemical Engineering 8 (2020) 104364”.

111.  In conclusion discuss the obtained scientific results.

112.  Add some updated literature and references.

Author Response

Thank you for your letter and reviewer's comments on our manuscript entitled " catalysts-1774462 ". Those comments are valuable and very helpful for revising and improving our paper, as well as the important guiding significance to our research. We have studied comments carefully and have made correction which we hope to meet with approval. Revised portion are marked in red in the manuscript. The main corrections in the manuscript and the responses to the reviewers' comments are as follows:

Reviewer #1

  1. The manuscript contains grammatical and typing mistakes which should be carefully read and remove and the language need improvement.

Response: Thank you for your comment. All mistakes in language have been modified as required. The details are as follows:the statement in line 13 (now in line 15) changes from “The catalysts were characterized by BET, EDS, SEM and XRD” to “BET, EDS and XRD characterized the catalysts”. In line 30 (now in line 35), change “as an” to “for”. In line 33 (now in line 35), change “around the world” to “worldwide”. In line 37 (now in line 42), change “a large number of” to “many”. In line 44 (now in line 52), change “the development of” to “developing”. From line 58 to 59 (now from line 65 to 67), change “To overcome these drawbacks, researchers fixed Fe2+(Fe3+) or iron oxides (FeII and FeIII) in Fenton reagent on the carrier to form a heterogeneous Fenton system” to “Researchers fixed Fe2+(Fe3+) or iron oxides (FeII and FeIII) in the Fenton reagent on the carrier to form a heterogeneous Fenton system to overcome these drawbacks”. From line 73 to 74 (now from line 81 to 82), change “and found that Cu-doped SBA-15 with Cu/Si mole ratio of 0.133 had the best catalytic activity for oxidative degradation of 4-chlorophenol” to “They found that Cu-doped SBA-15 with Cu/Si mole ratio of 0.133 had the best catalytic activity for oxidative degradation of 4-chlorophenol”. From line 88 to 89 (now from line 100 to 101), change “The electronegativity of Si-OH and the loss of coordination water in attapulgite are formed by the rupture of the Si-O bond” to “The rupture of the Si-O bond forms the electronegativity of Si-OH and the loss of coordination water in attapulgite”. In line 149 (now in line 163), change “the Congo red and methylene blue in the solution” to “the solution's Congo red and methylene blue”. In line 182 (now in line 181), change “the nitrogen adsorption and desorption isotherms of the catalyst” to “the catalyst's nitrogen adsorption and desorption isotherms”. In line 179 (now in line 197), change “rest of” to “remaining”. In line 231 (now in line 252), change “as well as” to “and”. In line 242 (now in line 263), change “be eventually” to “eventually be”. In line 247 (now in line 268), change “the amount of reactive free radicals generated by the reaction” to “the reaction's amount of reactive free radicals”. In line 301 (now in line 322), change “a great influence on” to “greatly influences”. In line 323 (now in line 344), change “RhB molecule was first adsorbed by the granular catalyst” to “the granular catalyst first adsorbed RhB molecule”. From line 328 to 330 (now from line 348 to 350), change “and the mineralization effect of dye was better in acidic conditions than in alkaline conditions, because H2O2 was relatively stable in acidic condition” to “The mineralization effect of dye was better in acidic conditions than in alkaline conditions, because H2O2 was relatively stable in acidic conditions”. In line 388 (now in line 408), change “of” to “in”. In line 419 (now in line 440), change “the surface of the catalyst” to “catalyst's surface”.  From line 432 to 433 (now from line 449 to 450), change “Through the degradation experiment of different dyes, the feasibility of its application in dye wastewater can be better clarified” to “The feasibility of its application in dye wastewater can be better clarified through the degradation experiment of different dyes” From line 508 to 509 (now from line 525 to 526), change “a small part of RhB molecules were catalyzed by hydroxyl radicals in the liquid phase.” to “hydroxyl radicals catalyzed a small part of RhB molecules in the liquid phase.” From line 551 to 552 (now from line 567 to 569), change “generation amount of hydroxyl radical in the liquid phase increased first and then decreased, indicating that the higher the H2O2 concentration” to “hydroxyl radical generation in the liquid phase increased first and then decreased, indicating that the higher the H2O2 concentration”. In line 565 and 569 (now in line 582 and 586), change “the surface of the catalyst” to “catalyst's surface”. In line 581 (now in line 598), change “Mechanism of catalytic oxidation with high iron is as follows” to “The mechanism of catalytic oxidation with high iron is as follows”.

  1. Discuss some toxic effect of Rhodamine B dye in the introduction section of the manuscript.

Response: Thanks for your correction. Some toxic effects of the target contaminant rhodamine B have been added in the introduction section, from lines 44 to 47.

  1. Provide the maximum absorbance of methylene blue and congo red dyes.

Response: Thank you for your comment. The maximum absorbance of methylene blue and Congo red dyes was 664nm and 497nm, respectively.

  1. Why Figure 1b is discussed before than 1a. Arrange in series.

Response: Thanks for your correction. In the compilation of the article, there was a mistake in the typesetting of the article, which led to inconsistent discussion of the two figures. The discussions in Figure 1a and Figure 1b have been changed in the article.

  1. 5. Provide JCDPF file no. to all the XRD of all the samples.

Response: Thanks for your correction. The XRD JCDPF file number was 87-2096. The XRD peaks of the samples have been labeled and analyzed. XRD spectra of iron-loaded catalysts shows that attapulgite characteristic diffraction peaks appeared at 2θ=8.5°, 19.8°, 20.3°, 27.9°, 35.7°, etc. The corresponding crystal plane spacing was D (110) =10.3974 nm, D (040) =4.4725 nm, D (021) =4.3829 nm, D (400) =3.1942 nm, D (002) =2.5139 nm, respectively. The SiO2 characteristic diffraction peaks appeared at 2θ=26.7°, 36.6° and 50.2°, and the corresponding crystal plane distances were D (011) =3.3421 nm, D (110) =2.4560 nm, D (011) =1.8171 nm, respectively. In addition, α-Fe2O3 cubic spinel characteristic diffraction peaks were found at 2θ=33.2°, 35.6°, 49.5°, 54.1°, etc. The corresponding crystal plane spacing was D (104) =2.6989 nm, D (110) =2.5171 nm, D (024) =1.8408 nm, D (116) =1.6943 nm, but the overall peak strength was much weaker than Fe2O3 crystal.

  1. Perform SEM and TEM characterization for surface analysis.

Response: Thanks for your correction. SEM analysis of granular catalysts has been supplemented in section 3.1.2. However, TEM analysis of granular catalyst was not carried out in this experiment, because the granular catalyst was made of attapulgite rubbed. According to the requirements of transmission electron microscope sample, electron beam could not penetrate the surface of granular catalyst, and the coarse cross section of granular catalyst was not suitable for such characterization.

  1. 7. Write catalysts sustainability instead of catalysts use.

Response: Thanks for your correction. In this paper, catalyst use has been changed to catalyst sustainability in line 64, 438 and 441.

  1. Label the corresponding peaks with crystal planes in the XRD figure.

Response: Thanks for your correction. The corresponding peaks have been marked with crystal planes in the XRD figure.

9.The authors should properly correlate the XRD data with the results presented in the manuscript.

Response: Thanks for your correction. According to the XRD pattern of the catalyst, only a small part of Fe (NO3)3 was converted into α-Fe2O3 crystal in the calcination process, while most of the rest of Fe (NO3)3 May be converted into other crystal iron oxides or only adhered to the surface of the catalyst in the form of some chemical bonds without showing the corresponding crystal structure. It can be used to discuss the repeatability and stability of the catalyst. The iron leaching rate of the catalyst is described in section 3.3, which is very low. High degradation rate can be achieved after repeated use for five times. Therefore, it also indicates that some Fe (NO3)3 May be converted into other crystalline iron oxides on the surface of the granular catalyst or merely attached in the form of some chemical bonds.

  1. Explain the mechanism and affecting parameters with the help from these reviews and cite in the manuscript “Water 2022, 14, 242. https://doi.org/10.3390/w14020242 and Journal of Environmental Chemical Engineering 8 (2020) 104364”.

Response: Thanks for your guidance. References 68 and 69 are of great help to the study of the mechanism of dye degradation by granular catalyst in this paper, which can better explain how catalyst catalyzes dye degradation, and further speculate the mechanism of dye degradation by granular catalyst.

  1. In conclusion discuss the obtained scientific results.

Response: Thanks for your guidance. Some of the results are discussed in the conclusion, and some important results of this study are added.

  1. Add some updated literature and references.

Response: Thanks for your correction. Some new literature has been added to the article. For example, references 7, 17 and 18, and 68 and 69. These literatures are helpful to better explore the process and mechanism of catalytic degradation of pollutants by catalysts.

Reviewer 2 Report

This manuscript (catalysts-1774462) presents application heterogeneous Fenton based AOPs system to remove Rh B from aqueous solution. The effect of experimental conditions was studied, and optimum parameters were selected.  Attapulgite particles were loaded with iron ions, and the resulting iron-bearing attapulgite particle catalyst was used to degrade RhB dye. The fundamental properties of granular catalyst have been investigated. To my opinion, I believe that the subject of this investigation is suitable for publication in Journal of Cleaner Production and the theme is interesting. Therefore, I recommend some revisions of the manuscript before publication and hope to improve it:

My comments:

1.    I suggest the authors to rewrite the abstract with a focus on background, objectives, methodology, main findings and conclusion. Please add a sentence which shows the necessity of the study

2.    The quality of figures and figure captions should be improved. Error bars should be added to all experimental data.

3.    The introduction part is not well structured for the paragraphs need to be revised to match the centre topic of this manuscript and relevant references should be cited to support paragraphs. Meanwhile, there is no enough description of innovation and scientific issues in this part. It is to be suggested that the author should clearly indicate the aim & scope of the paper; it is to be mentioned how the study is useful for the theoretical and practical purposes. More information to improve the introduction section can be introduced:

https://doi.org/10.1016/j.psep.2017.11.005   

doi:10.1088/1755-1315/958/1/012011

4.    Rewrite the research hypothesis (last paragraph of the introduction section)

5.    The Discussion is too simple and not deep enough. More close connections and quantitative results should be established and discussed. See the references. The major weakness of the manuscript is the poor discussion of the findings. Please describes the findings and relate them to the literature.

6.    Kinetic investigation is missing.

7.    The conclusion can be reduced in length to be more informative.

8.     Authors are recommended to revise the whole manuscript for the language proof.

Author Response

Thank you for your letter and reviewer's comments on our manuscript entitled " catalysts-1774462 ". Those comments are valuable and very helpful for revising and improving our paper, as well as the important guiding significance to our research. We have studied comments carefully and have made correction which we hope to meet with approval. Revised portion are marked in red in the manuscript. The main corrections in the manuscript and the responses to the reviewers' comments are as follows:

Reviewer #2:

  1. I suggest the authors to rewrite the abstract with a focus on background, objectives, methodology, main findings and conclusion. Please add a sentence which shows the necessity of the study

Response: Thanks for your correction. I have rewritten the abstract to include the research background, objectives, methods, major findings and conclusions, and the necessity of the research. 

  1. The quality of figures and figure captions should be improved. Error bars should be added to all experimental data.

Response: Thank you for your comment. The quality of figures and numerical instructions in the article has improved.  All experimental data were not parallel tested without error bar because the materials were made by hand and difficult to keep same. 

  1. The introduction part is not well structured for the paragraphs need to be revised to match the centre topic of this manuscript and relevant references should be cited to support paragraphs. Meanwhile, there is no enough description of innovation and scientific issues in this part. It is to be suggested that the author should clearly indicate the aim & scope of the paper; it is to be mentioned how the study is useful for the theoretical and practical purposes.

Response: Thanks for your correction. The paragraph structure of the introduction is modified, and references 6 and 14 are cited to support the central theme of the article, so that the theme of the article is more prominent. The purpose and scope of this paper are added in the thesis. In lines 88-92, we add a description of this.

  1. Rewrite the research hypothesis (last paragraph of the introduction section

Response: Thanks for your correction. The research hypothesis has been rewritten and some statements have been modified.

  1. The Discussion is too simple and not deep enough. More close connections and quantitative results should be established and discussed. See the references. The major weakness of the manuscript is the poor discussion of the findings. Please describes the findings and relate them to the literature.

Response: Thanks for your guidance. Studies have found that under the condition of pollutant concentration changing, the number of molecules per unit volume increases, while the amount of reactive free radical generation is constant, so the degradation and mineralization rate of RhB decreases with the increase of concentration [1]. At the same time, the amount of active free radicals in the heterogeneous Fenton system also changed with the change of hydrogen peroxide concentration. At a certain amount of hydrogen peroxide, the active free radicals produced in the system can effectively attack RhB molecules. When the concentration exceeds a certain level, hydrogen peroxide may produce its own extinction effect and thus the active free radicals produced will play an ineffective role. Under the condition of high catalytic dose, the adsorption site and reactive site of RhB molecule increased, and the adsorption and degradation of RhB molecule were played, and the number of oxidized free radicals in the system was increased. At the same time, the pH of the reaction solution should not be too high or too low, because the over-acid and over-alkali environment will destroy the structure of the granular catalyst, resulting in the dissolution of particles and the fall off of iron ions. Too high pH will inhibit the formation of •OH [2], while too low pH of the reaction solution will destroy the conversion balance between Fe â…² and Fe â…± on the catalyst surface, affecting the catalytic reaction [3]. The generation of active free radicals requires energy, so the higher the temperature in the system, the higher the energy [4], the more likely the O—O bond of H2O2 is to break, and the higher the concentration of active free radicals, which greatly accelerates the oxidative degradation process [5]. In the process of reactive degradation of RhB, the n-site deethyl and large conjugated oxanthracene structures of RhB molecule were destroyed by hydroxyl and peroxyradicals.

  1. Kinetic investigation is missing.

Response: Thank you for your comment. In this paper, the dynamic changes of rhodamine B catalyzed by granular catalyst were studied. The UV-VIS spectra of rhodamine B, the determination of active substances and the formation of main active substances in the reaction process were studied. Kinetics of degradation of rhodamine B by granular catalyst has not been comprehensively studied, and this part has been adjusted in the abstract.

  1. The conclusion can be reduced in length to be more informative.

Response: Thank you for your comment. The conclusion has been shortened and some information has been added. (From lines 613 to 637)

  1. Authors are recommended to revise the whole manuscript for the language proof.

Response: Thank you for your comment. All mistakes in language have been modified as required. The details are as follows:the statement in line 13 (now in line 15) changes from “The catalysts were characterized by BET, EDS, SEM and XRD.” to “BET, EDS and XRD characterized the catalysts”. In line 30 (now in line 35), change “as an” to “for”. In line 33 (now in line 38), change “around the world” to “worldwide”. In line 37 (now in line 42), change “a large number of” to “many”. In line 44 (now in line 52), change “the development of” to “developing”. From line 58 to 59 (now from line 65 to 67), change “To overcome these drawbacks, researchers fixed Fe2+(Fe3+) or iron oxides (FeII and FeIII) in Fenton reagent on the carrier to form a heterogeneous Fenton system” to “Researchers fixed Fe2+(Fe3+) or iron oxides (FeII and FeIII) in the Fenton reagent on the carrier to form a heterogeneous Fenton system to overcome these drawbacks”. From line 73 to 74 (now from line 81 to 82), change “and found that Cu-doped SBA-15 with Cu/Si mole ratio of 0.133 had the best catalytic activity for oxidative degradation of 4-chlorophenol” to “They found that Cu-doped SBA-15 with Cu/Si mole ratio of 0.133 had the best catalytic activity for oxidative degradation of 4-chlorophenol”. From line 88 to 89 (now from line 100 to 101), change “The electronegativity of Si-OH and the loss of coordination water in attapulgite are formed by the rupture of the Si-O bond” to “The rupture of the Si-O bond forms the electronegativity of Si-OH and the loss of coordination water in attapulgite”. In line 149 (now in line 163), change “the Congo red and methylene blue in the solution” to “the solution's Congo red and methylene blue”. In line 182 (now in line 181), change “the nitrogen adsorption and desorption isotherms of the catalyst” to “the catalyst's nitrogen adsorption and desorption isotherms”. In line 179 (now in line 197), change “rest of” to “remaining”. In line 231 (now in line 252), change “as well as” to “and”. In line 242 (now in line 263), change “be eventually” to “eventually be”. In line 247 (now in line 268), change “the amount of reactive free radicals generated by the reaction” to “the reaction's amount of reactive free radicals”. In line 301 (now in line 321), change “a great influence on” to “greatly influences”. In line 323 (now in line 344), change “RhB molecule was first adsorbed by the granular catalyst” to “the granular catalyst first adsorbed RhB molecule”. From line 328 to 330 (now from line 348 to 350), change “and the mineralization effect of dye was better in acidic conditions than in alkaline conditions, because H2O2 was relatively stable in acidic condition” to “The mineralization effect of dye was better in acidic conditions than in alkaline conditions, because H2O2 was relatively stable in acidic conditions”. In line 388 (now in line 408), change “of” to “in”. In line 419 (now in line 440), change “the surface of the catalyst” to “catalyst's surface”.  From line 432 to 433 (now from line 449 to 450), change “Through the degradation experiment of different dyes, the feasibility of its application in dye wastewater can be better clarified” to “The feasibility of its application in dye wastewater can be better clarified through the degradation experiment of different dyes” From line 508 to 509 (now from line 525 to 526), change “a small part of RhB molecules were catalyzed by hydroxyl radicals in the liquid phase.” to “hydroxyl radicals catalyzed a small part of RhB molecules in the liquid phase.” From line 551 to 552 (now from line 567 to 569), change “generation amount of hydroxyl radical in the liquid phase increased first and then decreased, indicating that the higher the H2O2 concentration” to “hydroxyl radical generation in the liquid phase increased first and then decreased, indicating that the higher the H2O2 concentration”. In line 565 and 569 (now in line 582 and 586), change “the surface of the catalyst” to “catalyst's surface”. In line 581 (now in line 598), change “Mechanism of catalytic oxidation with high iron is as follows” to “The mechanism of catalytic oxidation with high iron is as follows”.

Reference

[1] Bai, C., Gong, W., Feng, D., Xian, M., Zhou, Q., Chen, S., Ge, Z., Zhou, Y., 2012. Natural graphite tailings as heterogeneous Fenton catalysts for the decolorization of rhodamine B. Chem. Eng. J. 197:306-313. https://doi.org/10.1016/j.cej.2012.04.108

[2] Wang, C., Cao, Y., Wang, H., 2019. Copper-based catalyst from waste printed circuit boards for effective Fenton-like discoloration of Rhodamine B at neutral pH. Chemosphere. 230:278-285. https://doi.org/10.1016/j.chemosphere.2019.05.068

[3] Krysa, J., Mantzavinos, D., Pichat, P., Poulios, I., 2018. Advanced oxidation processes for water and wastewater treatment. Environ. Sci. Pollut. Res. 25(35): 34799-34800. https://doi.org/10.1007/s11356-018-3411-2

[4] Feng, J., Hu, X., Yue, P., Qiao, S., 2009. Photo Fenton degradation of high concentration Orange II (2mM) using catalysts containing Fe: A comparative study. Sep. Purif. Technol. 67(2): 213-217. https://doi.org/10.1016/j.seppur.2009.03.013

[5] Hashemian, S., 2013. Fenton-like oxidation of malachite green solutions:Kinetic and thermodynamic study. J. Chem. https://doi.org/10.1155/ 2013/809318

Reviewer 3 Report

This paper describes the “synthesis of attapulgite supported iron catalyst and its application to degradation of dye in wastewater”. I recommend major revision based on the following concerns.

1.        The authors should describe the novelty of their research, especially in the abstract and conclusion.

2.        Many same kinds of catalysts have been mentioned in literature. What is the difference between the current understudy catalyst to others already mentioned in the literature? The authors should provide a table comparing the efficiency of their catalysts with those already reported in the literature.

3.        Why authors selected Rhodamine B as an example for dye decomposition? What are the side effects of the release of this dye in wastewaters? The authors should highlight the importance of this study of dye decomposition in the paper.

4.        The paper should be rewritten according to the scientific standards using scientific terms. For example, “Attapulgite support” is written as the “carrier”. All such mistakes should be revised.

5.        The English language of the paper needs extensive editing.

6.        The title of the paper should be more precise.

7.        The authors should include the following highly related and important references.

https://doi.org/10.1016/j.jenvman.2016.05.075

https://doi.org/10.1246/bcsj.20150052

Author Response

Thank you for your letter and reviewer's comments on our manuscript entitled " catalysts-1774462 ". Those comments are valuable and very helpful for revising and improving our paper, as well as the important guiding significance to our research. We have studied comments carefully and have made correction which we hope to meet with approval. Revised portion are marked in red in the manuscript. The main corrections in the manuscript and the responses to the reviewers' comments are as follows:

Reviewer #3

  1. The authors should describe the novelty of their research, especially in the abstract and conclusion.

Response: Thanks for your guidance. The abstract and conclusion have been revised to supplement the innovation of this paper.

  1. Many same kinds of catalysts have been mentioned in literature. What is the difference between the current understudy catalyst to others already mentioned in the literature? The authors should provide a table comparing the efficiency of their catalysts with those already reported in the literature.

Response: Thanks for your correction. The catalyst in this paper was prepared from natural clay minerals with low cost and abundant yield. With large specific surface area, unique spatial crystal structure and ion exchange on the surface, it can not only be used as catalyst carrier for catalytic degradation of pollutants, but also has certain adsorption capacity, and has good adsorption capacity for organic pollutants in water. Therefore, it provides a certain practical value and industrial prospect for cheap natural materials. Compared with other catalysts, granular catalyst has large contact area, less iron leaching and higher reuse rate. Comparison of different types of catalyst for catalytic degradation efficiency, on the basis of different preparation conditions, processing pollutant environment, different impact conditions and different pollutants, these factors will lead to different types of catalyst for catalytic degradation efficiency is different, so only through narrative of different types of catalyst under the best conditions for the degradation efficiency of pollutants.

  1. Why authors selected Rhodamine B as an example for dye decomposition? What are the side effects of the release of this dye in wastewaters? The authors should highlight the importance of this study of dye decomposition in the paper.

Response: Thanks for your correction. Many organic dyes were toxic and carcinogenic, causing significant damage to human health and the environment [1]. In addition, due to its complex structure [2], it was difficult to degrade in the water environment. Rhodamine B was a flavane and highly water-soluble red dye with high toxicity [3]. Due to its low price, rhodamine B was widely used in textile and food industry [4]. In addition to skin staining, this dye may also cause various diseases, such as cancer [5]. Therefore, rhodamine B was selected as the target pollutant in this paper. After the dye wastewater is discharged into the water, it will cause the color change of the water body, consume the dissolved oxygen in the water, and be enriched by the organisms in the water, and be absorbed by the human body through the food chain, causing great harm to the human body.

  1. The paper should be rewritten according to the scientific standards using scientific terms. For example, “Attapulgite support” is written as the “carrier”. All such mistakes should be revised.

Response: Thanks for your correction. The paper has been revised and rewritten in scientific terms according to scientific standards. In line 12 (now in line 15), change “Attapulgite support” to “carrier”. In line 30 (now in line 35), change “as an” to “for”. In line 33 (now in line 38), change “around the world” to “worldwide”. In line 37 (now in line 42), change “a large number of” to “many”. In line 44 (now in line 52), change “the development of” to “developing”. In line 96 (now in line 108), change “precipitation” to “the precipitation”. In line 179 (now in line 197), change “rest of” to “remaining”. In line 231 (now in line 252), change “as well as” to “and”. In line 301 (now in line 322), change “a great influence on” to “greatly influences”. In line 397 (now in line 418), change “the addition of” to “adding”.

  1. The English language of the paper needs extensive editing.

Response: Thanks for your correction. All mistakes in language have been modified as required. The details are as follows:the statement in line 13 (now in line 15) changes from “The catalysts were characterized by BET, EDS, SEM and XRD.” to “BET, EDS and XRD characterized the catalysts”. In line 30 (now in line 35), change “as an” to “for”. In line 33 (now in line 38), change “around the world” to “worldwide”. In line 37 (now in line 42), change “a large number of” to “many”. In line 44 (now in line 52), change “the development of” to “developing”. From line 58 to 59 (now from line 65 to 67), change “To overcome these drawbacks, researchers fixed Fe2+(Fe3+) or iron oxides (FeII and FeIII) in Fenton reagent on the carrier to form a heterogeneous Fenton system” to “Researchers fixed Fe2+(Fe3+) or iron oxides (FeII and FeIII) in the Fenton reagent on the carrier to form a heterogeneous Fenton system to overcome these drawbacks”. From line 73 to 74 (now from line 81 to 82), change “and found that Cu-doped SBA-15 with Cu/Si mole ratio of 0.133 had the best catalytic activity for oxidative degradation of 4-chlorophenol” to “They found that Cu-doped SBA-15 with Cu/Si mole ratio of 0.133 had the best catalytic activity for oxidative degradation of 4-chlorophenol”. From line 88 to 89 (now from line 100 to 101), change “The electronegativity of Si-OH and the loss of coordination water in attapulgite are formed by the rupture of the Si-O bond” to “The rupture of the Si-O bond forms the electronegativity of Si-OH and the loss of coordination water in attapulgite”. In line 149 (now in line 163), change “the Congo red and methylene blue in the solution” to “the solution's Congo red and methylene blue”. In line 182 (now in line 181), change “the nitrogen adsorption and desorption isotherms of the catalyst” to “the catalyst's nitrogen adsorption and desorption isotherms”. In line 179 (now in line 197), change “rest of” to “remaining”. In line 231 (now in line 252), change “as well as” to “and”. In line 242 (now in line 263), change “be eventually” to “eventually be”. In line 247 (now in line 268), change “the amount of reactive free radicals generated by the reaction” to “the reaction's amount of reactive free radicals”. In line 301 (now in line 321), change “a great influence on” to “greatly influences”. In line 323 (now in line 344), change “RhB molecule was first adsorbed by the granular catalyst” to “the granular catalyst first adsorbed RhB molecule”. From line 328 to 330 (now from line 348 to 350), change “and the mineralization effect of dye was better in acidic conditions than in alkaline conditions, because H2O2 was relatively stable in acidic condition” to “The mineralization effect of dye was better in acidic conditions than in alkaline conditions, because H2O2 was relatively stable in acidic conditions”. In line 388 (now in line 408), change “of” to “in”. In line 419 (now in line 440), change “the surface of the catalyst” to “catalyst's surface”. From line 432 to 433 (now from line 449 to 450), change “Through the degradation experiment of different dyes, the feasibility of its application in dye wastewater can be better clarified” to “The feasibility of its application in dye wastewater can be better clarified through the degradation experiment of different dyes” From line 508 to 509 (now from line 525 to 526), change “a small part of RhB molecules were catalyzed by hydroxyl radicals in the liquid phase.” to “hydroxyl radicals catalyzed a small part of RhB molecules in the liquid phase.” From line 551 to 552 (now from line 567 to 569), change “generation amount of hydroxyl radical in the liquid phase increased first and then decreased, indicating that the higher the H2O2 concentration” to “hydroxyl radical generation in the liquid phase increased first and then decreased, indicating that the higher the H2O2 concentration”. In line 565 and 569 (now in line 582 and 586), change “the surface of the catalyst” to “catalyst's surface”. In line 581 (now in line 598), change “Mechanism of catalytic oxidation with high iron is as follows” to “The mechanism of catalytic oxidation with high iron is as follows”.

  1. The title of the paper should be more precise.

Response: Thanks for your correction. The original title was “Study on degradation characteristics and mechanism of dye wastewater by iron-bearing attapulgite granular catalyst”, now the title was “Degradation of rhodamine B in wastewater by iron-loaded attapulgite particle heterogeneous Fenton catalyst”.

  1. The authors should include the following highly related and important references.

Response: Thanks for your guidance. Some highly relevant and important references have been added to the paper. For example, reference 7, 15, 67, etc., these references are helpful to understand the toxicity of dyes and the process and mechanism of catalytic degradation of dyes.

Reference

[1] Kumbhar, S., S, Mahadik, M., A, Shinde, S., S, Rajpure, K., Y, Bhosale, C., H, 2015. Fabrication of ZnFe2O4 films and its application in photoelectrocatalytic degradation of salicylic acid. J. Photochem. Photobiol. B: Biol. 142:118–123. https://doi. org/10.1016/j.jphotobiol.2014.12.002

[2] Li, G., Wang, Y., Mao, L., 2014. Recent progress in highly efficient Ag-based visible-light photocatalysts. RSC Adv. 4: 53649–53661. https://doi.org/10.1039/C4RA08044K

[3] Jain, R., Mathur, M., Sikarwar, S., Mittal, A., 2007. Removal of the hazardous dye rhodamine B through photocatalytic and adsorption treatments. J. Environ. Manag. 85:956–964. https://doi.org/10.1016/j.jenvman.2006.11.002

[4] Natarajan, T., S, Thomas, M., Natarajan, K. Bajaj, H., C, Tavade, R., J, 2011. Study on UV-LED/TiO2 process for degradation of Rhodamine B dye. Chem. Eng. J. 169:126–134. https://doi.org/10.1016/j.cej.2011.02.066

[5] Serbian, I., Hoenke, S., Kraft, O., Csuk, R., 2020. Ester and amide derivatives of rhodamine B exert cytotoxic effects on different human tumor cell lines. Med. Chem. Res. 29:1655–1661. https://doi.org/10.1007/s00044-020-02591-8

Round 2

Reviewer 1 Report

Most of the changes have incorporated but still need some revisions, which are

1. The labelling in the XRD figure is not clear, the figure need to be redrawn with high quality.

2. Provide the JCPDS file no in the mansucript.

3. Provide the maximum absorbance peaks of methylene blue and congo red in the manuscript.

4. Use sustainibility only in the heading as 3.3 Catalysts sustainibility. In the text use the word reuse instead of sustainibility

Author Response

Thank you for your letter and reviewer's comments on our manuscript entitled " catalysts-1774462 ". Those comments are valuable and very helpful for revising and improving our paper, as well as the important guiding significance to our research. We have studied comments carefully and have made correction which we hope to meet with approval. Revised portion are marked in red in the manuscript. The main corrections in the manuscript and the responses to the reviewers' comments are as follows:

Reviewer #1

  1. The labelling in the XRD figure is not clear, the figure need to be redrawn with high quality.

Response: Thank you for your comment. XRD patterns have been redrawn and relabeled.

  1. 2. Provide the JCPDS file no in the mansucript.

Response: Thanks for your correction. The JCPDS file number has been added at line 157 (now in line 147).

  1. 3. Provide the maximum absorbance peaks of methylene blue and congo red in the manuscript.

Response: Thanks for your correction. In line 164 (in line 154), The maximum absorption wavelengths of Congo red and methylene blue added were 497nm and 664nm respectively.

  1. Use sustainibility only in the heading as “3.3 Catalysts sustainibility”. In the text use the word “reuse” instead of “sustainibility”.

Response: Thanks for your correction. In line 64, 437 and 440 (now in line 58, 412 and 415), change “sustainability” to “reuse”. In line 404 (now in line 381), change “reuse” to “sustainability”.

Reviewer 2 Report

The authors have revised the manuscript well according to the reviewers' comments. I think that the paper could be accepted for publication.

Reviewer 3 Report

The authors revised the manuscript following the comments.

For comment No. 4, the authors misunderstood. In the manuscript, the authors mentioned 

“Attapulgite carrier” which is not appropriate and should be changed to “Attapulgite support”.

Author Response

Thank you for your letter and reviewer's comments on our manuscript entitled " catalysts-1774462 ". Those comments are valuable and very helpful for revising and improving our paper, as well as the important guiding significance to our research. We have studied comments carefully and have made correction which we hope to meet with approval. Revised portion are marked in red in the manuscript. The main corrections in the manuscript and the responses to the reviewers' comments are as follows:

Reviewer #3

  1. For comment No. 4, the authors misunderstood. In the manuscript, the authors mentioned 

“Attapulgite carrier” which is not appropriate and should be changed to “Attapulgite support”.

Response: Thanks for your correction. The paper has been revised and rewritten in scientific terms according to scientific standards. In line 15 (now in line 11), change “Attapulgite carrier” to “Attapulgite support”. In line 68 (now in line 61), change “carriers” to “supports”. In line 601 (now in line 570), change “carrier” to “supported”. Thank you again for your advice in your busy schedule.